# AI Framework for Generative Design of Computational Experiments with Structures in Physical Environment

**Gleb V. Solovev**
NSS Lab, ITMO University
Saint-Petersburg, Russia
glebsolo46@gmail.com

**Anna Kalyuzhnaya**
NSS Lab, ITMO University
Saint-Petersburg, Russia
anna.kalyuzhnaya@itmo.ru

**Alexander Hvatov**
NSS Lab, ITMO University
Saint-Petersburg, Russia
alex_hvatov@itmo.ru

**Nikita O. Starodubcev**
NSS Lab, ITMO University
Saint-Petersburg, Russia
nstarodubtcev@itmo.ru

**Oleg Petrov**
NSS Lab, ITMO University
Saint-Petersburg, Russia
ogpetrov@itmo.ru

**Nikolay O. Nikitin**
NSS Lab, ITMO University
Saint-Petersburg, Russia
nnikitin@itmo.ru

## Abstract

We discuss the applicability of an open-source generative design for the automated design of computational experiments with structures in physical environments for various scientific fields. It may be used for scientific experiments where the searched structure can be represented as a set of 2D non-oriented graphs with any topology (grids, polygons, trees), and the physical environment can be described with any numerical model (classic or data-driven). The proposed framework gives the tools to efficiently explore a space of experiment configurations with generative AI models and evolutionary algorithms. The results are shown in examples from different fields: design of microfluidic devices, coastal engineering, research on heat transfer, and acoustics. Due to the framework's focus on working with structures as graphs, it is possible to pre-train generative NN that is used to create an initial population of optimized structures. The framework finds application in diverse areas such as coastal engineering, acoustics, engineering design, heat transfer, hydrodynamics, and medicine.

## 1 Introduction

In the current age, AI undoubtedly has the potential to be an essential tool to generate and test scientific hypotheses in various fields of knowledge(1). Assistance for scientific search with an end-to-end AI modeling approach is in high demand for fields with well-developed numerical modeling (e.g., in physics (2) or chemistry (3)), as it helps to get results faster and better. In mechanical design, several tools perform experiments without human intervention and incorporate active learning to choose subsequent experiments (e.g., the BEAR tool(4), which uses Bayesian networks(5)). In recent years, in materials and molecular sciences, a new paradigm – closed-loop discovery – utilizes inverse design and automates a whole workflow to enable faster identification, scale-up, and manufacturing (6). As well as ideas and methods for close-loop automated laboratory research(7).

Examples of successful applications of generative design, artificial intelligence (AI), and machine learning (ML) are numerous in many fields (8; 9). In medicine, some tasks require the analysis and processing of many variants. Drug creation is often lengthy and expensive, but generative design can speed up this crucial aspect of healthcare. Using AI algorithms, scientists can generate and evaluate millions of potential drug candidates, optimizing factors such as efficacy, safety, and pharmacokinetics

(10). In addition, neural networks diagnose diseases (such as cancer). The method is based on the analysis of the deformability of cells. Since it is expensive to perform such measurements manually, CNN simplifies the research and increases its precision (11).

As another example of generative methods, a large language model based on the design of novel sequences of spider silk proteins to meet complex combinations of targeted mechanical properties (12). Deep-learning approaches also extend the existing knowledge of metamaterial design in acoustics, even for materials with unique structures that are quite difficult for humans to develop. Through periodically arranged structures, acoustic metamaterials can influence sound propagation in acoustic media. This study significantly expands our understanding of design strategies for sound insulation tasks (13). There are also examples from other fields (14; 15).

The existing results demonstrate that AI-assisted design of scientific experiments is achievable and is in demand by researchers in different fields. Along with local successes and confirmation of hypotheses about the applicability of AI methods, there is a lack of tools for universalizing the process of experiment optimization and automation, at least for some classes of experiments. The paper proposes an open-source framework GEFEST[1] (16) for the automated generative design of computational experiments with geometrically encoded structures in physical environments for various scientific fields. This tool can help researchers automate their experiments by reducing search space by representing target solutions as geometrical structures in the presence of some numerical simulator.

The workflow for scientific experiments usually combines the real-life and computer-based (in-silico) stages. The proposed framework focuses on integrating generative design methods and algorithms to discover the preliminary set of optimal structures and reduce search space in the virtual part of the experiments. The problem statement for this task is described in Section 2.

We provide examples of the applicability of a generative approach for experimental cases in different fields. The high-level design of the coastal breakwaters is generated before the stages of full-scale modeling and preparation of the construction project (see Subsection 4.2. The same is true for micro-scale: before the manufacturing of various configuration blood cell traps, the set of preliminary solutions can be generated using optimization techniques and generative models (see Subsection 4.1). Similar cases are considered for acoustic and thermodynamic experiments (Subsection 4.4 and Subsection 4.3).

The domain-agnostic implementation of the proposed approach is available as an open-source framework GEFEST. It can be configured to simplify the computer-based part of experiments in different fields of applied science. It is described in Section 3. The future development of this research will be focused on extensive use of pretrained generative models and transfer of learning between scientific fields to reduce the computational cost of the proposed approach, as described in Section 5.

## 2   Problem statement

The human-based design process in experimental research sequentially includes designing a prototype, conducting a computer simulation of the object to assess its physical qualities, and analyzing the results. Depending on the result's quality, this chain of actions is repeated until the required qualities of the designed object are achieved. This approach takes considerable time and requires considerable knowledge from the expert (this sequence of experimentation is demonstrated in Figure 1 A)). It is challenging to optimize experiments in this setting. Often, a considerable number of variants of different structures are required, which is much more complicated if the characteristics of the desired object are evaluated by several criteria (for example, when designing breakwaters, it is necessary not only to consider the effectiveness of wave height reduction but also to understand how expensive the designed configuration will be in construction).

Applying the framework in the science research lifecycle allows one to reduce the participation of expert engineers in experimental research (an example of optimized experiment lifecycle is demonstrated in Figure 1 B). That is achieved because the automated cycle of designing objects and evaluating their characteristics is performed without human participation. Then, thanks to evolutionary optimization algorithms, the best options are selected and improved in the next cycle.

---

[1]https://github.com/aimclub/GEFEST

As a result, a human is only required to set input parameters and constraints for the desired objects and analyze the best prototypes created by the framework.

In addition, because the framework and computational computing can be easily scaled using more computing power, it is possible to speed up experiments significantly. In addition, the ability to use surrogate samplers and surrogate estimators(17) with physics-based loss functions(18) and models-simulators of the sensitivity analysis of physical processes allows the experimental process to be significantly accelerated.

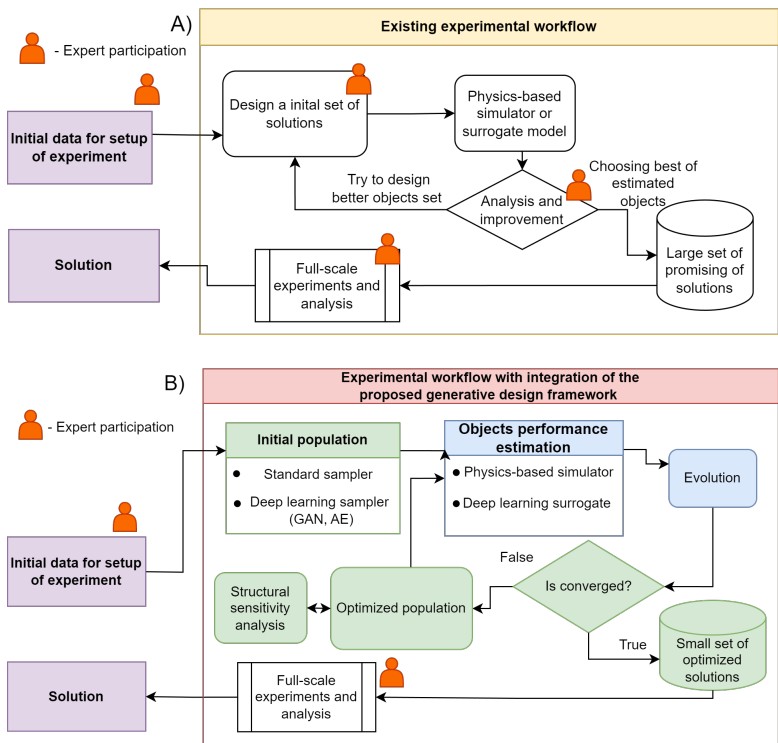

Figure 1: Research workflows: (A) existing workflow; (B) workflow integrated with GEFEST generative design framework.

# 3   Proposed solution

The proposed approach combines deep learning, numerical simulation, and evolutionary optimization to generate and improve the two-dimensional representation of natural objects in a given domain. The workflow includes polygon coding, toolbox creation, and generative design itself. Figure 2 illustrates the main modules of the framework.

The main stage of the proposed workflow is the generative design. Its implementation is detailed in Figure 2 and the Algorithm 1.

Generative design is based on three steps: sampling, estimation, and optimization. All three steps can be combined differently depending on the problem to be solved. For example, if we need to find a solution quickly, we can skip the optimization step (lines 1, 3, 4, 11, and 12) and conduct *random search*. If the quality of solutions is essential, a *traditional approach* is carried out: Lines 1, 3, 4, and 6 of the 1 algorithm. A single sampling operation is performed, and evaluation and optimization are repeated until the stopping criterion is reached. Therefore, this process has an inherent low exploration speed. However, increasing the speed of exploration in some tasks is necessary. That can be done by combining lines 8 and 9, that is, by performing an additional sampling operation at each step of the loop; let us call it *extra sampling*.

**Algorithm 1** Generative design of setup for computational experiment

**Require:** $P = (S, E, O, SA)$            ▷ User-defined toolkit
**Ensure:** $D_{fin}$            ▷ Final designed objects
  1: $D_{curr} \leftarrow S.\text{sample}()$            ▷ Initial designs
  2: **while** $stopCriteria$ **do**
  3:      $PF \leftarrow E.\text{estimate}(D_{curr})$            ▷ Design performance
  4:      $D_{curr} \leftarrow E.\text{select}(PF, D_{curr})$            ▷ Selecting $k$ best objects
  5:      **if** $O.\text{required}$ **then**
  6:          $D_{curr} \leftarrow O.\text{optimize}(D_{curr}, PF)$
  7:          **if** $S.\text{required}$ **then**
  8:              $D_{sample} \leftarrow S.\text{sample}()$
  9:              $D_{curr} \leftarrow D_{sample} \cup D_{curr}$            ▷ Combination
10:          **if** $SA.\text{required}$ **then**
11:              $D_{curr} \leftarrow SA.\text{analyse}(D_{curr}, PF)$    ▷ Performing sensitivity analysis after a given number of optimisation iterations
12:      **else**            ▷ Skip optimization
13:          $D_{sample} \leftarrow S.\text{sample}()$
14:          $D_{curr} \leftarrow D_{sample} \cup D_{curr}$
15: $D_{fin} \leftarrow D_{curr}$
16: **return** $D_{fin}$

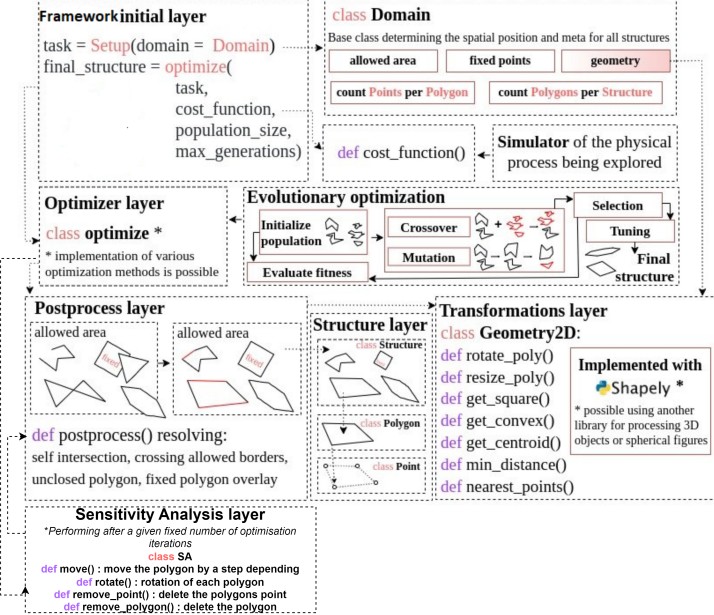

Figure 2: The proposed framework for the generative design of physical objects used in experiments.

# 4 Experimental studies

## 4.1 Microfluidics studies

Microfluidic experimental studies can be an appropriate candidate for applying generative design techniques. An example is developing a microfluidic trap with the unique function of trapping a single red blood cell (RBC) in flow channels by through-slit flow (19). In this problem, it was necessary to design a blood microfilter configuration. It is necessary to configure the design so that the bloodstream passing through the L-shaped "traps" reaches the necessary velocity to capture red blood cells.

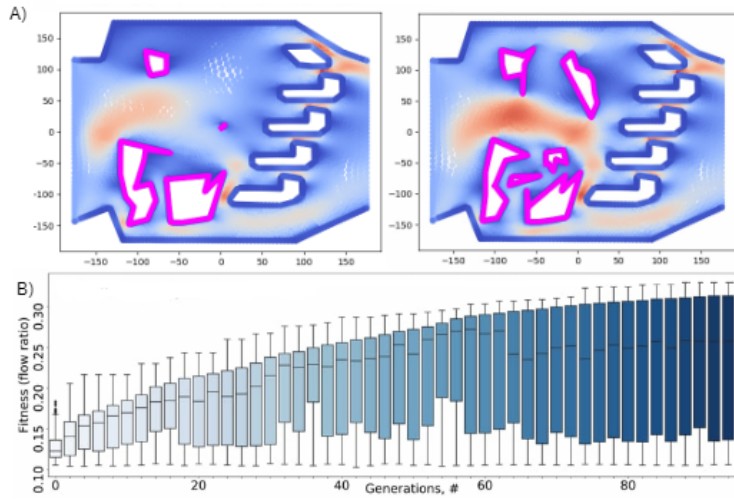

Figure 3: Generative design of microfluidic devices. A) The flow of particles passes through this device with different structures in the process of generative design. Barriers can be located inside the optimization domain. B) Convergence of evolutionary algorithm

The existing workflow is based on sequential modeling, fabrication, and validation. Since there are many potential solutions to this problem, finding the optimal solution will require much expert work and computational resources. In this problem, integrating generative design into the research process allows not only to achieve efficient values of the desired metric (this is shown in Figure 3B)) However, it also finds non-obvious solutions for humans, which are demonstrated by 3A).

The proposed approach partially solves this problem. After searching through 30000 solutions, an optimized geometry was found that increases blood speed flow through slit traps by 49%(20).

This task uses a deep learning-based generator for geometric objects (barriers in the blood filter). We use the architecture of adversarial autoencoder, which allows achieving a 75% reduction in object sampling time without loss in quality.

## 4.2   Coastal Engineering

Structural engineers are also often faced with tasks requiring multiple experiments. That is because many independent variables are selected from the reference ranges that engineers use in their calculations. Also important is the role of the expert's experience and imagination, which can allow them to anticipate the optimal solution. However, if the problem is unfamiliar to the engineer, reviewing all possible variants may take considerable time.

As an example of the experimental task in structural engineering, we consider the design of wave protection structures. In this field, the construction of real-world models (21) provides the closest results to the natural conditions, but the cost of testing many breakwater configurations is enormous. In this case, having the breakwater designed by an expert engineer and calculated in specialized software becomes optimal. It takes a lot of time and effort for an expert to analyze the applicability of candidate solutions and combine the best parts with the final one.

A baseline configuration was chosen to evaluate the framework's performance in this task. It was built manually, as an engineer could have chosen it in the first approximation (shown in Figure 4 (a)). When evaluated by the SWAN simulator (22), the baseline configuration had a height reduction in the protected zone at three points to 0.21, 0.29, and 0.32 m. After performing the generative design with the combination of SWAN+surrogate solver, a solution was obtained with target wave height values of 0.02,0.16 and 0.23 m (optimized configuration shown in Figure 4 (b)). In addition, the neural surrogate model reduces computational time. From this, it is concluded that the proposed generative design framework can be applied in automated coastal defense studies.

Table 1: Comparison between proposed GD framework and Baseline configuration.

| Toolkit | Wave hights (m) | | |
|---|---|---|---|
| Proposed framework | 0.02 | 0.16 | 0.23 |
| Baseline | 0.21 | 0.29 | 0.32 |
| Improvement by % | 95 | 55.1 | 71.8 |
| Mean improvement by % | 74 | | |

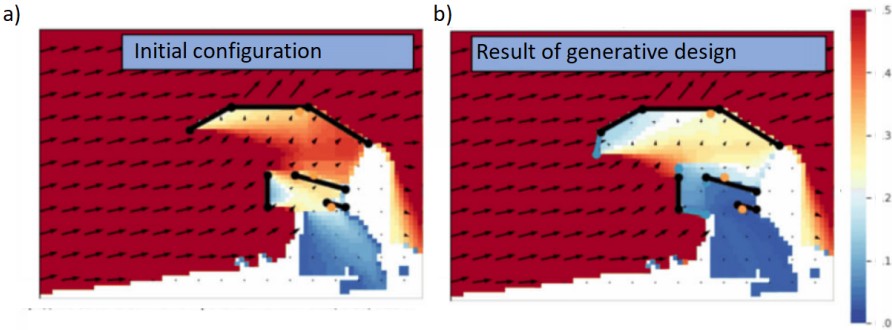

Figure 4: The results of simulation for different structures of breakwaters: (a) initial setup; (b) setup improved by generative design (the breakwater link after optimization is shown in blue).

## 4.3 Thermal management

The experimental pipeline in thermodynamics and heat transfer can also be improved using generative design. In the paper, we consider the problem of heat management, which involves optimizing the temperature distribution inside a microdevice. It is presented in Figure 5 a).

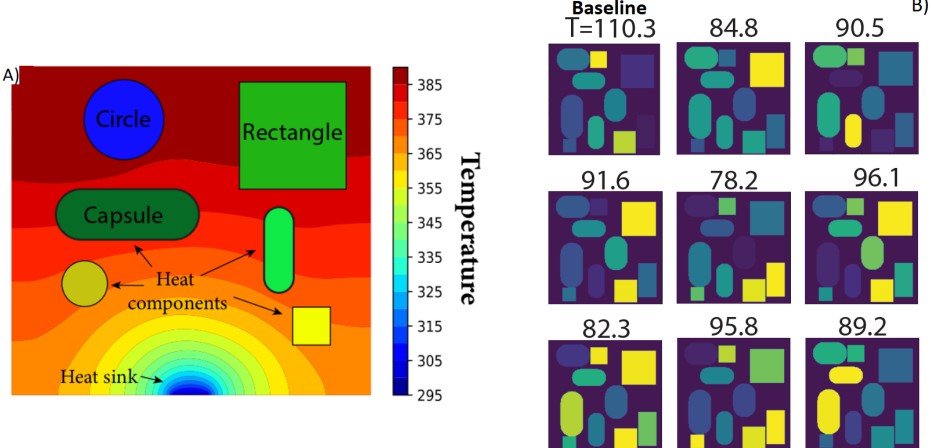

Figure 5: Result for experiments with heat source placement: (a) Configuration of the designed heat source. It consists of three types of heat components and a heat sink; (b) The improved configurations obtained with deep learning.

The iterative modification of the heat sources for each new chip to find a satisfactory solution can be time-consuming and costly. At the same time, there are open datasets for heat source placement (23). So, we trained deep generative models and avoided direct simulator runs of new simulations. Then,

this model was used to evaluate the fitness during the optimization stage for a specific microdevice. It makes it possible to reduce the initial temperature to 29% (the results are presented in Figure 5 b).

## 4.4 Signal processing

An essential task in signal processing is the detection and reconstruction of the shape of objects from the sound signal interacting with these objects. For example, such tasks are investigated in flaw detection and echolocation. We have considered a simplified case of the problem of object shape reconstruction from observations of the surrounding sound field. In the framework of the experiment, cases with different numbers of sound receivers (sensors) were investigated.

In the real-world stage of experiments, the use of many sensors is associated with significant financial costs. In addition, placing the sensors in space and processing the measurements at each location is time-consuming. The involvement of generative design allows us to empirically evaluate the relationship between the number of receivers and the quality of object shape reconstruction. That allows us to estimate how many receivers are needed for satisfactory object shape reconstruction. Our experiments show that a satisfactory result can be obtained using 64 receivers (the location of receivers is marked by red asterisks in the figure Figure 6). That can be seen from Table 2, which shows how many individuals in the population have reached a certain level of the Dice coefficient (0.85 and 0.95). The table shows that the metric does not even reach the value of 0.85 when measuring with 9 receivers. However, when measuring with 64 receivers, the metric reaches 0.95 (almost complete matching of the reconstructed object area) for 15% of the object population.

Table 2: Experimental results for baseline (BL) and proposed evolutionary algorithm (Evo) for various number of iterations for optimisation. MAE and Dice criterion are involved.

| Receivers | Exp | Loss per iterations (std+/- max/min) | | | | Percentile of population individs with Dice more than 0.85/0.95 | | | |
|---|---|---|---|---|---|---|---|---|---|
| | | 50 | 100 | 150 | 200 | 50 | 100 | 150 | 200 |
| 9 | BL | $0.17^{+0.07}_{-0.07}$ | $0.15^{+0.04}_{-0.06}$ | $0.13^{+0.04}_{-0.04}$ | $0.13^{+0.04}_{-0.04}$ | 0/0 | 0/0 | 0.11/0 | 0.11/0 |
| | **Evo** | $\mathbf{0.12^{+0,09}_{-0,06}}$ | $\mathbf{0.1^{+0.04}_{-0,04}}$ | $\mathbf{0.08^{+0.04}_{-0,03}}$ | $\mathbf{0.07^{+0.04}_{-0,03}}$ | 0/0 | 0/0 | 0/0 | 0/0 |
| 64 | BL | $0.33^{+0.06}_{-0.05}$ | $0.3^{+0.05}_{-0.03}$ | $0.3^{+0.05}_{-0,02}$ | $0.29^{+0.06}_{-0.04}$ | 0/0 | 0/0 | 0/0 | 0.22/0 |
| | **Evo** | $\mathbf{0.18^{+0.06}_{-0,09}}$ | $\mathbf{0.15^{+0.09}_{-0,07}}$ | $\mathbf{0.13^{+0.11}_{-0,09}}$ | $\mathbf{0.11^{+0.12}_{-0,08}}$ | **0.15/ 0** | **0.23/ 0** | **0.30/ 0.07** | **0.46/ 0.15** |
| 240 | BL | $0.26^{+0.14}_{-0.07}$ | $0.25^{+0.08}_{-0.06}$ | $0.23^{+0.05}_{-0.04}$ | $0.22^{+0.05}_{-0.06}$ | 0.22/ 0 | 0.22/ 0 | 0.22/ 0 | 0.22/ 0 |
| | **Evo** | $\mathbf{0.17^{+0.1}_{-0.07}}$ | $\mathbf{0.14^{+0.09}_{-0.05}}$ | $\mathbf{0.11^{+0.11}_{-0.07}}$ | $\mathbf{0.1^{+0.08}_{0.06}}$ | **0.30/ 0** | **0.54/ 0.07** | **0.54/ 0.15** | **0.69/ 0.23** |
| Full field | BL | $0.41^{+0.06}_{-0.07}$ | $0.39^{+0.07}_{-0.05}$ | $0.36^{+0.03}_{-0.02}$ | $0.35^{+0.04}_{-0.04}$ | 0/0 | 0/0 | 0.22/0 | 0.44/0 |
| | **Evo** | $\mathbf{0.21^{+0.16}_{-0.17}}$ | $\mathbf{0.14^{+0.22}_{-0,1}}$ | $\mathbf{0.11^{+0.2}_{-0,06}}$ | $\mathbf{0.08^{+0.11}_{-0,05}}$ | **1.0/ 0.15** | **1.0/ 0.30** | **1.0/ 0.54** | **1.0/ 0.69** |

Also, Figure 7 shows boxplots with loss function $MAE$, Dice coefficient, and a custom metric to evaluate the quality of object shape reconstruction depending on the number of receivers called $Reconstruction$. To understand that the object shape is reconstructed qualitatively, it is necessary that the loss function decreases and the Dice score grows. The $Reconstruction$-metric shows it by its growth.

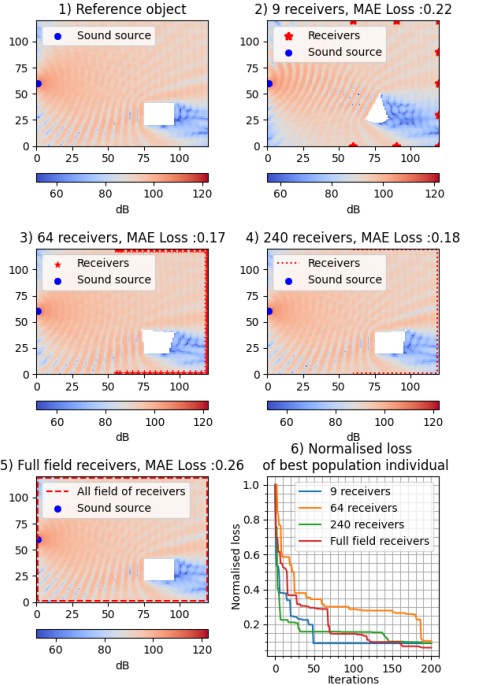

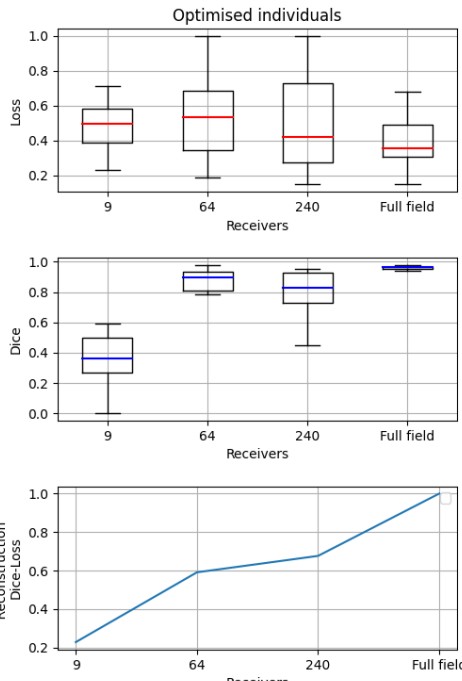

Figure 6: Results of reconstruction defects with rectangular shapes. The figure presented the reference object shape (1) and examples of shape reconstruction using different numbers of receivers (9, 64, 240, and the Full field).

Figure 7: Dependence of object shape reconstruction quality on the number of receivers.

Our proposed approach streamlines the experimental program, making assessing various iterations of the problem formulation easier. Additionally, if the final step of the experiments involves conducting tests in real-world settings, generative design enables the optimal execution of these tests. In this instance, we determined the minimum number of receivers required for the experiment since the uncertainty of object shape restoration is high, even when a numerical model is directly available.

## 5 Conclusion

In the paper, we demonstrate that the integration of generative design with a workflow of experiments offers a helpful tool for scientists in multiple fields. This framework can enhance scientific innovation, optimize performance, and facilitate data-driven decision-making by enabling efficient exploration of experiment configurations and leveraging generative AI models.

The paper proposes a generative framework for conducting automated experiments and optimizing the preliminary stages of experiment design. This approach is based on creating a random population of desired objects, estimating their natural qualities, and evolutionary optimization. Generative design is actively applied in many fields of science and is a promising area for optimizing research and experimentation. Examples of problems solved by the considered framework show its advantages in universality and the possibility of application in entirely different areas of physical sciences related to continuous media.

We demonstrated the effectiveness of the proposed approach for DoE optimization for different case studies and improved initial solutions from 8% to 74% depending on the domain and tasks. While it already provides a helpful instrument for AI-based experiments, future extensions of the approach are required. First of all, the involvement of pre-trained generative models can reduce the computational cost of the optimization; the automated discovery of lightweight models allows for avoiding evaluations of physics-based simulators and even makes it possible to produce the initial variants of experimental setups in zero-shot mode. Finally, the meta-learning can utilize the experience from previous experiments in the same field.

# 6 Acknowledgements

This work was supported by the Analytical Center for the Government of the Russian Federation (IGK 000000D730321P5Q0002), agreement No. 70-2021-00141.

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
