# OpenReview forum: "AI Framework for Generative Design of Computational Experiments with Structures in Physical Environment"
_NeurIPS.cc/2023/Workshop/AI4Science — NeurIPS2023-AI4Science Poster_

### Official Review · Reviewer_JcDz · 2023-10-06
**Review for AI Framework for Generative Design of Computational Experiments with Structures in Physical Environment**

**Rating:** 3
**Confidence:** 3

**Review:**

## Summary
Existing experimental workflows requires designing prototypes, conducting computer simulations, and analysing the results until a required quality of a designed object is achieved. This approach is considerable time consuming and requires expert-knowledge.
Here, the authors represents a framework, which automates the cycle of designing objects and evaluating their characteristics without human intervention. The human needs only needs to set input parameters and constraints and analyse the best constructed prototypes.

## Clarity
The paper suffers from issues of readability that could be significantly improved. Specifically, enhanced guidance for the reader such as introducing figures and their relevance before they appear could make a substantial difference. Also guide the reader through the figures, what is shown in them and include labels on the figures. Similarly, the method that is presented is nearly not described: What is the physics-based simulators, the deep learning surrogate (including loss function, early stop criteria, architecture etc.), what is the conversion criteria used for the experiments etc.

Minor comment: Line 148: Is it Fig. 5 that is referred to?

## Evaluation of the quality
It is extremely difficult to judge the quality of the work as the work is poorly-described.

However, in general the result sections lacks baseline results.

## Originality and significance
It might be that the authors have made an algorithm, which can optimize microfluidic trap geometries, structures of breakwaters, the configurations of microdevices or object shape reconstruction in signal processing. However, I think that the paper needs substantial work to clarify this. It is not clear what the algorithm does,  how to interpret the results and these are not compared properly with baselines.

I hope the authors can use my comments to improve the papers.

Best of luck :)

---

### Official Review · Reviewer_S9Cq · 2023-10-24
**Generative Design Applied to Various Physical Systems**

**Rating:** 7
**Confidence:** 3

**Review:**

The authors propose a generative design with three steps: sampling, estimation, and optimization, and apply it to microfluidics, wave protection structures, heat sources and sound signalling. The authors demonstrate that the proposed framework can effectively improve performance. The proposed framework is well described, and I suggest accepting it.

The authors only compare the proposed framework with a baseline workflow that is non-generative. Also as the authors mentioned, generative designs are widely or at least actively used in many fields of science for optimizing research and experimentation. How is this framework different from other generative designs? I would like to learn more about the comparison of performance between this framework and other generative frameworks.

Another weakness is the lack of details about the specific models used in each example, which makes it hard to evaluate the reproducibility.

---

### Meta-Review · Area_Chair_EmQ6 · 2023-10-27

**Recommendation:** Accept (Poster)
**Confidence:** 2

**Metareview:**

The authors describe a general approach for generative design of prototypes for various types of experiments. The paper is somewhat difficult to evaluate and could benefit from significant revision before final submission: problem formulations and figure legends are somewhat vague, stronger generative field-specific baselines should be used, methods should be explicitly described for reproducibility, and evaluations should be more thorough. The paper is relevant to the workshop, may be of interest in many areas, and could generate useful discussions.